# Safety of Clinical Ultrasound Neuromodulation

**DOI:** 10.3390/brainsci12101277

**Published:** 2022-09-22

**Authors:** Sonja Radjenovic, Gregor Dörl, Martin Gaal, Roland Beisteiner

**Affiliations:** Functional Brain Diagnostics and Therapy, Department of Neurology, Medical University of Vienna, Spitalgasse 23, A-1090 Vienna, Austria

**Keywords:** transcranial ultrasound, neuromodulation, TPS, FUS, clinical neuromodulation

## Abstract

Transcranial ultrasound holds much potential as a safe, non-invasive modality for navigated neuromodulation, with low-intensity focused ultrasound (FUS) and transcranial pulse stimulation (TPS) representing the two main modalities. While neuroscientific and preclinical applications have received much interest, clinical applications are still relatively scarce. For safety considerations, the current literature is largely based on guidelines for ultrasound imaging that uses various physical parameters to describe the ultrasound pulse form and expected bioeffects. However, the safety situation for neuromodulation is inherently different. This article provides an overview of relevant ultrasound parameters with a focus on bioeffects relevant for safe clinical applications. Further, a retrospective analysis of safety data for clinical TPS applications in patients is presented.

## 1. Introduction

Over the past decade, there have been considerable advances in developing new sonication methods for non-invasive brain stimulation [1,2,3]. Although deep brain stimulation (DBS) is known as an optimistic treatment method for various clinical indications including epilepsy, essential tremor, dystonia, and Parkinson’s disease [4], it is a surgical procedure, and the risk of side-effects cannot be completely ruled out [5]. After treatment with Gamma Knife (GK) thalamotomy, reduced tremor and improvement in everyday activities was seen in patients. However, the effects are usually delayed, and bilateral tremor could not be improved with this treatment method [5]. The growing interest in ultrasound-based modalities over electrical stimulation techniques, such as transcranial magnetic stimulation [6] or transcranial direct current stimulation [7], is due to a number of advantages. These include, for instance, the high spatial accuracy [8] or possibility of non-invasive subcortical stimulation [9], which have been limitations for electrical stimulation techniques, especially in clinical settings [10,11]. While these characteristics apply in general to ultrasonic neuromodulation, notable varieties in available transducer systems produce different stimulation pulses. Low-intensity focused ultrasound (FUS) systems generate sonication trains of sinus tones with a fixed fundamental frequency that is usually (though not necessarily) applied in a pulsed fashion [12]. Next to FUS, transcranial pulse stimulation (TPS) represents another ultrasound-based stimulation technique with markedly different characteristics. TPS generates ultrashort pressure pulses consisting of multiple frequencies with higher amplitude [13]. The pulse duration is in the range of a few microseconds, and typically administered with a repetition frequency between 1 and 5 Hz, reducing the risk of tissue heating associated with continuous application of ultrasound [14]. Both FUS and TPS can be described with a number of interacting parameters that can be controlled to change the physical properties of the stimulation pulse. Among these are, for instance, the fundamental frequency (typically between 250–1000 kHz), pulse repetition frequency (determining the pulse rate), duty cycle (the percentage of the time the ultrasound pulse is “on” over the smallest stimulation period) and the intensity.

Transcranial ultrasound neuromodulation has been extensively investigated in pre-clinical settings, both in animal and human studies, yet clinical applications are still scarce. The first uses were in patients with disorders of consciousness, both with a non-navigated TPS precursor [15] and navigated FUS [16]. Although neither trial had sham controls, results were promising and showed no signs of neurological damage. The first clinical study with highly focused navigated pulses has been performed with the TPS technology [13]. In this multicenter clinical study with Alzheimer patients, TPS was shown to be safe and specifically improve cognition, memory and depression scores [13,17,18]. Later, navigated FUS studies investigated patients with disorders of consciousness [19,20], Alzheimer’s disease [21] and epilepsy [22]. While the possible clinical applications for precise neural stimulation are immense, it is vital to deepen our understanding of the exact mechanisms and consequences of brain sonication in neural tissue to offer safe and effective treatment options. 

Clinical ultrasound has the potential to produce two major types of effects with relevance for clinical safety, namely heating and cavitation [23]. These modalities strongly differ from others. Tissue heating occurs due to the absorption of the ultrasonic waves, and heating increases with ultrasound frequency and the applied acoustic intensity [14]. Cavitation relates to the expansion and collapse of local tissue exposed to a tensile pressure. If local gas bodies exist, harming effects on local tissue cells may increase. In order to avoid related bioeffects during ultrasonic imaging, regulation authorities use indices to establish guidelines regarding the maximal output parameters of medical ultrasonic devices. Although there are clinical ultrasound imaging guidelines, there is a need to evaluate and decide on safety parameters specifically tailored to ultrasound application modalities, as the respective goals and, therefore, the physical requirements are quite different. The goal of this article is to provide an overview of relevant safety parameters and points of consideration for the clinical use of transcranial ultrasound stimulation, as well as to present results on the safe application in human subjects. 

## 2. Main Bioeffects

### 2.1. Cavitation Effects

The Mechanical Index (MI) was developed to adequately indicate the likelihood of cavitation [24]. The formula for the MI is based on the assumption that a nucleation site of only the resonant site is available for inertial cavitation [23]:(1)$$MI=\frac{P_{r}}{\sqrt{f_{c}}}$$
with *P_r_* being the derated peak rarefactional pressure [MPa] and *f_c_* the center frequency of the ultrasound pulse [MHz]. Notably, the MI is not applicable for TPS pressure pulses since the timescale of the expansion phase of a bubble forced by a TPS pulse is much longer than the TPS pulse length [25]. Concerning conditions for occurrence of cavitation in biological tissues, a wide range of cavitation thresholds are reported in the literature, demonstrating that this phenomenon largely depends on exposure and experimental conditions. The findings from several studies looking at cavitation effects are summarized below. One study investigated the cavitation threshold in sheep brains exposed to 660 kHz ultrasonic pulses (2 cycles) [25]. No cavitation could be detected for peak tensile pressures below 11.6 MPa, but systematically at 22.4 MPa. Another study found a reversible opening of the blood brain barrier (BBB) in rats after 50 focused shockwaves with tensile pressures of 9.8 MPa [26]. With the prior adjunction of microbubbles, BBB disruption and FUS induced cavitation using 20 ms bursts at 1.5 MHz at peak tensile pressures around 0.5 MPa were reported [27]. In the investigation of cavitation thresholds induced by ultrasound pulses (1–2 cycles) with a 1.1 MHz transducer in different media [28], a cavitation threshold (50% probability) from 14 to 30 MPa tensile pressure, depending on the sonicated medium, was produced. Other studies reported cavitation onset in a tissue mimicking phantom (agar) at 1 MHz for peak tensile pressures between 3 and 10 MPa [29,30]. The threshold for cavitation onset depended on the number of cycles (minimum 10 cycles), the duration of sonication (up to 3 s) and previous exposures. While researchers studying cavitation thresholds for 1–2 cycle histotripsy pulses at frequencies from 300 kHz to 3 MHz in water, phantoms and ex vivo bovine tissues [31] reported cavitation onset (50% probability) for peak tensile pressures from 24 to 30.6 MPa. In another study using a 1 MHz transducer delivering 5 histotripsy pulses repeated at a frequency of 100 to 1000 Hz, the same researchers reported initiation of dense bubble clouds in ex vivo porcine tissues from 1.5 (in lungs) up to 27 MPa (in other soft tissues) peak tensile pressures. They noted the importance of the tissue stiffness and mentioned the need for single bubbles to expand to a sufficient size during the initial cycles of the pulse in order to initiate a dense bubble cloud, as well as the reduction of the cavitation threshold at higher PRFs [32].

This variability in conditions and pressure ranges for generation of cavitation indicates that the MI alone is not a sufficient predictor of bioeffects [33,34]. Beyond the MI, other conditions have a major influence, such as the number of consecutive pulses applied, burst duration, or use of ultrasound contrast agents (UCA). The FDA guidelines limit the MI to 1.9 for diagnostic ultrasound devices [35]. No adverse non-thermal bioeffects have been observed for MI under the FDA limit in tissues without gas bodies, and the lowest threshold for cavitation in vivo and related adverse effects more likely lies at or above MI values of 4 [35].

The presence of injected or endogenous gas bodies in the sonicated medium is a critical condition for the onset of acoustic cavitation. Several authors reported a dramatic reduction in cavitation thresholds in organs naturally containing air bubbles, such as lungs and intestine, and in other tissues after systemic injection of UCA [34]. In brain injuries in rats following shockwave exposure from 7 up to 14 MPa tensile pressures after injection of UCA, a higher threshold was observed when no UCA was administered [36]. Another study reported only minor cavitation injuries in tissues exposed to clinically relevant lithotripsy exposures (>40 Mpa) [37]. However, this threshold decreased to 2 MPa in tissues containing endogenous gas bodies or when a UCA was injected.

### 2.2. Heating Effects

The energy deposit in tissues can be high enough to generate substantial local heating [14], which is desired for ablative indications such as MRI-guided focused ultrasound [38]. The local temperature rise depends on the tissue characteristics, such as absorption or perfusion. For instance, skull bone, due to its specific tissue properties, absorbs more acoustic energy and is subsequently more susceptible to heating [39]. The difference in temperature increase between skull and brain tissue was recently illustrated in a simulation using non-human primate data, where the authors showed an increase of 0.5 °C at the target site compared to 2.9 °C where the transducer was placed at the skull [40]. The Thermal Index (TI) was established to provide a reasonable estimate of temperature rise when ultrasound propagates through tissues, which is the ratio of the attenuated acoustic power on the acoustic power needed to raise the temperature by 1 °C at a specified point [41]. It is a unit-less value and has a recommended maximum of 6 in adults and should be adjusted according to the planned exposure time [42]. Another important physical metric related to thermal effects is the spatial-peak temporal average intensity (I_SPTA_), which gives the fraction of the sonication intensity per second, i.e., the time average over a continued pulse train: (2)$$I_{\mathrm{SPTA}}=\frac{1}{PRI}\;\int_{0}^{PRI}I_{SP}\left(t\right)dt$$
where *PRI* is the pulse repetition interval and *I_SP_* is the spatial average intensity [W/cm^2^]. This is related to the spatial-peak pulse average intensity (I_SPTA_), which gives the average over a single pulse:(3)$$I_{\mathrm{SPPA}}=\frac{1}{PD}\;\int_{0}^{PD}I_{SP}\left(t\right)dt$$
where *PD* is the pulse duration. Note that, depending on the mode of application, which is described by the fraction of sonication duration per second or duty cycle (DC), the I_SPTA_ will be different (pulsed application) or equal (continuous wave) to the I_SPPA_ [43]. Importantly, FDA guidelines for diagnostic imaging applications set the I_SPTA_ limit to 720 mW/cm^2^, which is often used as a reference, but this limit has been exceeded in past transcranial ultrasound stimulation studies without clinically relevant adverse events. 

### 2.3. Comparing Physical Properties of FUS and TPS

When comparing FUS and TPS procedures, therapeutic FUS generally consists of exposures to relatively low amplitude pressure waves in frequencies between 200 and 1000 kHz for a high number of consecutive cycles. Long bursts or continuous sonication are applied with durations up to several seconds. Although most FUS studies apply pressures <10 MPa, the I_SPTA_ involved typically reaches values of and above 1000 mW/cm^2^ ([23]; Table 3.2). In contrast, TPS pulses involve exposures to considerably higher peak pressures and use of short single pulses (a few microseconds) at very low repetition frequencies (1–10 Hz), where the I_SPPA_ below the skull may reach values around 4 kW/cm^2^. Considering the maximal I_STPA_ of TPS of 100 mW/cm^2^, and a corresponding TI << 0.7, the onset of thermal bioeffects in such conditions is very unlikely [42]. The produced intensity distribution is more focal compared with that of FUS [8]. A possible risk during long sonication trains using frequencies around 300 kHz is the unwanted generation of skull-conducted standing waves and subsequent secondary maxima [44,45]. This can be minimized when a pulsed FUS mode is used. Due to the short pulse lengths, for TPS no standing waves are expected.

## 3. Clinical Safety Data

### 3.1. Safety Assessments in Ovine, Porcine and Non-Human Primate Studies

The majority of studies in large animals and non-human primates have not reported any signs of thermal or mechanical damage due to ultrasound stimulation. Past studies in sheep, swine and non-human primates have also used I_SPTA_ or MI values exceeding FDA guidelines for imaging and reported only negligible rises in temperature [46,47,48,49]. One notable exception came in a sheep study [50], where the authors reported minor micro-hemorrhages in 4 sheep after stimulating with a very high repetition rate (≥500 times; 6.6–10.5 W/cm^2^ I_SPPA_, 0.9–1.2 MI). However, a later investigation using a range of comparable parameters did not find any signs of tissue damage after stimulation, arguing that the previous results were mainly due to preparation artefacts rather than mechanical damage from stimulation [51]. While the evidence remains inconclusive and warrants further investigation, this conflicting evidence illustrates the importance of methodological choice (as well as proper controls) in experimental designs when investigating thermal and mechanical effects of ultrasound stimulation. One long-term study investigated stimulation effects in 2 macaques over 2 years (129 and 147 stimulation sessions, respectively; 0.5–7.8 W/cm^2^ I_SPPA_ in water, 10.1–156.7 mW/cm^2^ I_SPTA_) and found no signs of trauma on MRI scans or in behavioral analysis [52].

### 3.2. Safety Assessments in Human Studies

Overall, ultrasound stimulation studies in humans have reported safe usage. Prudent safety evaluations in studies with human subjects are, for instance, structural MRI scans before and after stimulation sessions [13,53,54], neurological follow-up examinations [55] or, in a clinical setting, monitoring of physiological parameters during stimulation [19]. There has been only one histological analysis post-stimulation in humans reported to date. Stern et al. [56] sonicated areas planned for resection in patients with temporal lobe epilepsy with a range of 0.72–5.76 W/cm^2^ I_SPTA_. Tissue damage could be found in 1 out of 8 total patients; however, similar results were found in non-sonicated areas, indicating a possible preparation artefact.

In a comprehensive retrospective symptom report from 7 completed FUS neuromodulation studies in humans, which analyzed follow-up questionnaires of 64 study participants with self-perceived post-stimulation side-effects, among the participants, 7/64 reported mild-to-moderate symptoms that they felt were possibly or probably related to the stimulation, such as neck pain, difficulty paying attention, anxiety or muscle twitches. A subset of the study group received a second follow-up questionnaire 1–4 weeks after the first, in which no participant reported persisting or new symptoms. Notably, the positive symptom rate over all experiments correlated with an increase in I_SPPA_, indicating the importance of higher intensities to possible adverse events. While this study is limited due to other non-invasive stimulation methods being used in parts of their experiments (e.g., TMS), which are also known to cause side-effects, the authors show overall good tolerance to focused ultrasound stimulation.

In the first clinical investigation with navigated-focused stimulation performed with a total of 45 subjects (10 healthy, 35 Alzheimer patients, TPS technique), no major adverse events were reported [13]. Minor adverse events were noted with free reporting, namely headache (4%), mood deterioration (3%), pain (8%) and painless pressure sensations (17%) [57]. Further mild and transient adverse events that have been reported in ultrasound neuromodulation studies include scalp heating [22], feeling of pressure or slight pain at the stimulation site [53] or cognitive problems [22,58]. Importantly, similar adverse events have also been reported when sham stimulation was applied, indicating a notable placebo effect [59].

### 3.3. TPS Treatment Center Data

The TPS system NEUROLITH (Storz Medical AG, Tägerwilen, Switzerland) is the first clinically approved neuromodulation system based on ultrasound pulses. In a consecutive case series of TPS treatment requests at the clinical TPS Therapy and Development Center in Vienna, Austria, a retrospective analysis was performed. Clinical safety data from 101 patients (57.4% male, 42.6% female, 41-to-95 years old) suffering from neurodegenerative disease (e.g., dementia, Parkinson’s, and mixed and related disorders) were analyzed. The treatment concept corresponds to recommendations for personalized precision medicine with descriptions of individually optimized treatment plans. A treatment regime consisted of individualized 20–30 min TPS sessions per day for a minimum of two weeks. Depending on the individual pathology and network situation of the specific patient, treatment settings and targets were individually defined (brain areas, energy settings, TPS pulses per brain area, course of overall treatment). In total, 150–1500 pulses per brain area were administered corresponding to maximum values of about 0.1 W/cm^2^ I_SPTA_ and 4 kW/cm^2^ I_SPPA_. Patients underwent a structural MRI prior to the treatment (to rule out potential recent contraindications and allow optimized neuronal network planning). Every patient filled out questionnaires concerning adverse events after each treatment session. In total, 1010 questionnaires (corresponding to 10 sessions for 101 subjects) were collected and all completed forms were analyzed. Patients were asked to rate experienced pressure and headache during treatment on a scale from 0 to 10 (Table 1, Figure 1), as well as to freely describe adverse events during and after treatment sessions (Table 2, Figure 2). There were no follow-up questionnaires, limiting these results to only short-term risks.

Across all groups, over 80% of patients reported no sensation of pressure or pain during TPS treatment (Figure 1A). One patient reported notably higher pain sensation (7/10 in one session, 6/10 in two sessions) and one patient reported a higher-pressure sensation (8/10) in seven sessions (Table 1, Figure 2B). During TPS treatment, no major adverse events were reported, and the majority of patients completed the cycle without reporting any minor adverse events or dropping out. Minor adverse events included pressure (*n* = 6), pain (*n* = 4), an uncomfortable feeling (*n* = 4) or other sensations (4 patients, respectively, reported noise sensitivity, tingling, tenseness, heat sensation; Figure 2A,B). After treatment (each questionnaire was filled out at the following session), more than half of all patients reported no adverse events (Figure 2C,D). The most common adverse events were tiredness, dizziness and pain. Notably, if adverse events occurred, most patients reported these after only 1 out of 10 sessions. In the large majority of patients, clinical improvements were found in one or more clinical parameters; however, here we only present data on clinical safety and adverse events.

## 4. Conclusions

Ultrasonic brain stimulation shows much potential in a neuroscientific, translational and purely clinical context [3,60]. It is now possible to precisely target cortical as well as deep brain structures without targeting problems related to changed conductivity of pathological tissue [11]. Nevertheless, appropriate parameter settings are crucial for a safe as well as effective stimulation protocol. As this is still a relatively young field, further studies are needed to better understand bioeffects and achieve desired results with no risk of unintended damage. Concerning mechanical cavitation effects, the use of UCAs and more generally the presence of gaseous bodies within tissues seem to be the biggest risk for unwanted lesions. Heating risks are mainly related to high-intensity values, present in long sonication trains as opposed to single pulses. Current ultrasound stimulation modalities FUS and TPS operate within safe margins, with no relevant adverse events reported in human studies. For clinical applications, TPS has been shown to be safe, when sticking to our published patient protocols and exclusion/inclusion criteria, such as no signs of intracerebral trauma, no hemophilia or other blood clotting disease and no corticosteroid treatment within six weeks prior to TPS therapy [13]. The majority of patients noted no pressure or pain sensations during stimulation and only occasional and minor side-effects. While FUS still lacks comparable clinical data, previous reports in humans also indicate safe usage [19,58]. To ensure safety, monitoring before, during and after treatment is recommended. This can be achieved through MRI measurements immediately before treatment for exclusion of dangerous pathologies, EEG and fMRI recordings alongside therapy, questionnaires for patients about occurring side-effects, and reports on adverse events. To further advance the field, it is important to increase understanding of physical parameters and their influence on bioeffects in humans and clinical populations. 

## Figures and Tables

**Figure 1 brainsci-12-01277-f001:**
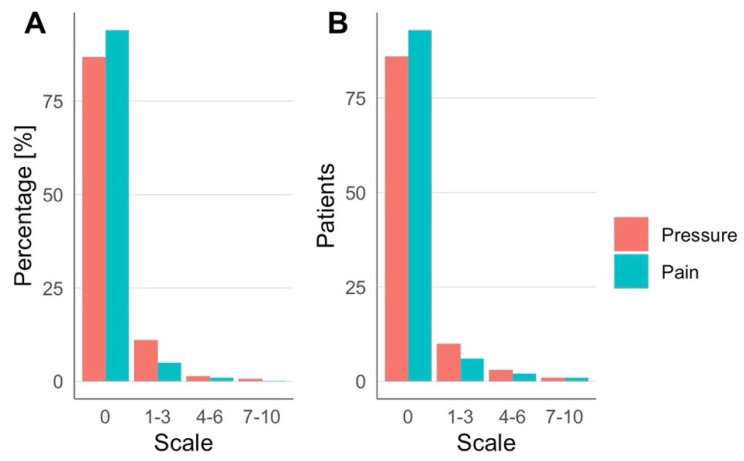
Experienced pressure and pain were rated during TPS treatment. (**A**) Percentages over all sessions and (**B**) number of patients across all sessions.

**Figure 2 brainsci-12-01277-f002:**
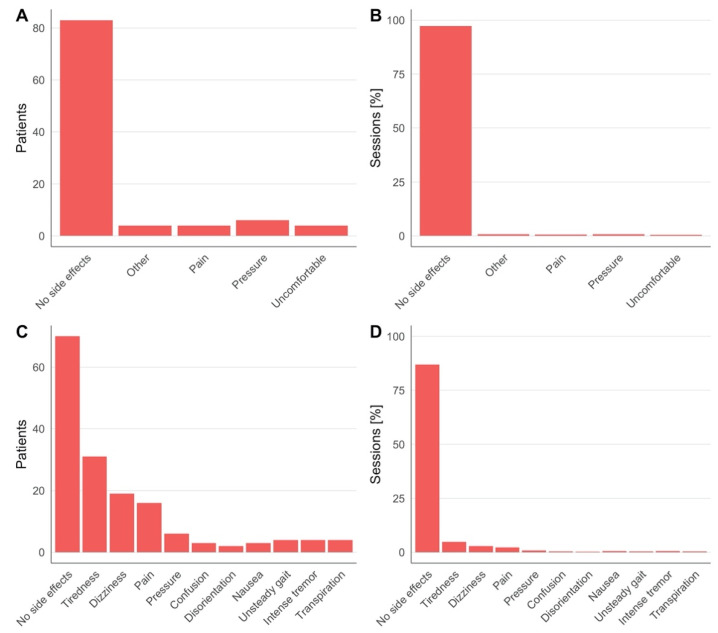
Adverse events during (**A**,**B**) and after (**C**,**D**). TPS treatment reported by patients across all sessions and percentage over all sessions, respectively. Detailed numbers are in Table 2.

**Table 1 brainsci-12-01277-t001:** Reported severity of pressure or pain sensation during TPS treatment. Scale on a range from 0–10. Data was taken from the TPS Therapy and Development Center in Vienna, Austria. Multiple scale occurrence was possible.

Scale	Pressure Number (%)	Pain Number (%)
Sessions	990	991
0	860 (86.67)	932 (94.05)
1–3	109 (11.01)	49 (4.94)
4–6	14 (1.41)	9 (0.91)
7–10	7 (0.71)	1 (0.1)
Patients	101	101
0	86	93
1–3	10	6
4–6	3	2
7–10	1	1

**Table 2 brainsci-12-01277-t002:** Adverse events reported during and after TPS treatment sessions (min. of 10 sessions per patient). Data was taken from all completed questionnaires, and multiple answers possible.

**Reported Adverse Events (during Treatment)**	**No. of Patients**	**No. of Sessions (%)**
No side-effects reported	83	966 (97.28)
Pressure	6	8 (0.81)

Pain	4	6 (0.6)
Uncomfortable	4	5 (0.5)
Other ^1^	4	8 (0.81)
**Reported Adverse Events (after Treatment)**	**No. of Patients**	**No. of Sessions (%)**
No side-effects reported	70	917 (86.92)
Tiredness	31	51 (4.83)
Dizziness	19	30 (2.84)
Pain	16	23 (2.18)
Pressure	6	9 (0.85)
Confusion	3	4 (0.38)
Disorientation	2	2 (0.19)
Nausea	3	5 (0.47)
Unsteady gait	4	4 (0.38)
Intensification of tremor	4	6 (0.57)
Sweating	4	4 (0.38)

^1^ Noise sensitivity, tingling, tenseness, heat sensation.

## Data Availability

The datasets analyzed for the current study are not publicly available due to patient confidentiality and participant privacy restrictions.

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
