# Peer review of "Safety of Clinical Ultrasound Neuromodulation"

_brainsci, 2022, doi:10.3390/brainsci12101277_

Round 1

Reviewer 1 Report

The role of ultrasound neuromodulation is an evolving feature of various entities, growing interest is associated with AD, PD, but also less common diseases as FXTAS. In this review authors elaborate on FUS and TPS. I have several comments which should be implemented in the text before further consideration:

1.     FUS and TPS are some of the neuromodulatory methods, however it would be valuable to discuss the highs and lows of the methods in comparison with other non-ultrasound methods as Deep Brain Stimulation or Gamma Knife

2.     The text would benefit, if authors could extended their discussion on the role of FUS depending on the diseases – more common as PD, ET, AD etc. and speculate on the feasibility as FXTAS – Ref. [1] Making a Difference-Positive Effect of Unilateral VIM Gamma Knife Thalamotomy in the Therapy of Tremor in Fragile X-Associated Tremor/Ataxia Syndrome (FXTAS). Front Neurol. 2018 Jun 27;9:512. doi: 10.3389/fneur.2018.00512. PMID: 29997574; PMCID: PMC6030249. [2] MRI-guided focused ultrasound thalamotomy in fragile X-associated tremor/ataxia syndrome. Neurology. 2016 Aug 16;87(7):736-8. doi: 10.1212/WNL.0000000000002982. Epub 2016 Jul 20. PMID: 27440151.

3.     The limitations of the method should be presented in a separate paragraph.

Author Response

POINT 1: FUS and TPS are some of the neuromodulatory methods, however it would be valuable to discuss the highs and lows of the methods in comparison with other non-ultrasound methods as Deep Brain Stimulation or Gamma Knife.

RESPONSE 1: We include a short comment on other non-ultrasound methods in the text (Lines 21-27).

POINT 2:  The text would benefit, if authors could extended their discussion on the role of FUS depending on the diseases – more common as PD, ET, AD etc. and speculate on the feasibility as FXTAS – Ref. [1] Making a Difference-Positive Effect of Unilateral VIM Gamma Knife Thalamotomy in the Therapy of Tremor in Fragile X-Associated Tremor/Ataxia Syndrome (FXTAS). Front Neurol. 2018 Jun 27;9:512. doi: 10.3389/fneur.2018.00512. PMID: 29997574; PMCID: PMC6030249. [2] MRI-guided focused ultrasound thalamotomy in fragile X-associated tremor/ataxia syndrome. Neurology. 2016 Aug 16;87(7):736-8. doi: 10.1212/WNL.0000000000002982. Epub 2016 Jul 20. PMID: 27440151.

RESPONSE 2: We include a comment on other methods for clinical neurology, such as Gamma Knife (Lines 21-27) and MR-guided focused ultrasound (Line 129). However, as we focused on non-invasive ultrasound neuromodulation a detailed discussion would go beyond the scope of this article.

POINT 3: The limitations of the method should be presented in a separate paragraph.

RESPONSE 3: We extend our discussion on some limitation (lines 276-279), however as limitations have been discussed elsewhere and the focus of our work is solely on safety considerations, a thorough discussion would go beyond this article.

Reviewer 2 Report

This study reviewed the safety and expected bioeffects of clinical ultrasound neuromodulation including low-intensity focused ultrasound (FUS) and transcranial pulse stimulation (TPS). The authors also reported a retrospective analysis of safety data for clinical TPS application. The structure is well organized and I find this manuscript very interesting. I have several minor suggestions which I hope will help the authors improve the manuscript.

1.    3.3. TPS treatment center data

More detailed patients characteristic would be helpful. For example, age, sex, medication status, diagnosis of neurodegenerative disease, and sonication site for the treatment.

2.    L224 said 90 patients data, and L229 and Tables/Figures said 101 subjects.

3.    Can the authors provide the summary table of reported adverse events in the sub-category such as age, sex, diagnosis, sonication site, etc.?  

4.    Since the authors summed reported adverse events at each time point and reported those in one table, we cannot tell short-term or immediate risks and long-term risks. It might be a good idea to report those events at each session from the initial session to 10 sessions, and >10 sessions, separately.

5.    Were there any dropouts from the sonication due to minor adverse events?

6.    What were the stimulation parameters in 3.3 TPS sessions (e.g., estimated tissue values for Isppa and Ispta) and sonication cycles in 30mins sonication protocols?

7.    Were those doses effective with patients? I see that this manuscript focuses on safety rather than efficacy; however, I am curious if those doses were enough to modulate the human brain to discuss safety with clinically meaningful doses.  

8.    L256. “Published patient protocols and published exclusion/inclusion criteria” Can the authors briefly summarize those protocols and exclusion/inclusion criteria in 3.3?

9.    Simultaneous EEG or fMRI recordings are additional potential approaches to monitoring safety issues in real-time.

10.  A summary statement of recommendation on how to set up a monitoring system for safety: before, during, and immediately after the sonication and in a long term, would be beneficial for readers.

Author Response

POINT 1: 3.3. TPS treatment center data

More detailed patients characteristic would be helpful. For example, age, sex, medication status, diagnosis of neurodegenerative disease, and sonication site for the treatment.

RESPONSE 1: The treatment concept corresponds to recommendations for personalized precision medicine with definition of individually optimized treatment plans. We now included more details, settings and patients’ characteristics (lines 229-239).

POINT 2: L224 said 90 patients data, and L229 and Tables/Figures said 101 subjects.

RESPONSE 2: This has been corrected (line 231).

POINT 3: Can the authors provide the summary table of reported adverse events in the sub-category such as age, sex, diagnosis, sonication site, etc.?  

RESPONSE 3: Due to the patient variability in this series of patient requested treatment attempts, subcategorization is not possible.

POINT 4: Since the authors summed reported adverse events at each time point and reported those in one table, we cannot tell short-term or immediate risks and long-term risks. It might be a good idea to report those events at each session from the initial session to 10 sessions, and >10 sessions, separately.

RESPONSE 4: We did not assess adverse events after the 10 session therapy cycle finished, therefore we cannot comment on adverse events after the treatment was finished. We include this in the text (lines 247-248).

POINT 5: Were there any dropouts from the sonication due to minor adverse events?

RESPONSE 5: There were no dropouts from TPS treatment. We include this in the text (line 259).

POINT 6: What were the stimulation parameters in 3.3 TPS sessions (e.g., estimated tissue values for Isppa and Ispta) and sonication cycles in 30mins sonication protocols?

RESPONSE 6: We added the information on maximum values in line 240.

POINT 7: Were those doses effective with patients? I see that this manuscript focuses on safety rather than efficacy; however, I am curious if those doses were enough to modulate the human brain to discuss safety with clinically meaningful doses.  

RESPONSE 7: We found clinical improvements in the majority of patients in one or more parameters, however we focus this article only on safety considerations. We extend our text with this point (lines 265-267).

POINT 8: L256. “Published patient protocols and published exclusion/inclusion criteria” Can the authors briefly summarize those protocols and exclusion/inclusion criteria in 3.3?

RESPONSE 8: We include this information in the text (lines 233-235; 284-287).

POINT 9: Simultaneous EEG or fMRI recordings are additional potential approaches to monitoring safety issues in real-time.

RESPONSE 9: We extend our discussion with this in the text (lines 292-293).

POINT 10: A summary statement of recommendation on how to set up a monitoring system for safety: before, during, and immediately after the sonication and in a long term, would be beneficial for readers.

RESPONSE 10: We include a summary with recommendations in the text (lines 290-293).

Round 2

Reviewer 1 Report

I don't have further comments.